# Landsat 9 Geometric Characteristics Using Underfly Data

**Michael J. Choate [1],\*, Rajagopalan Rengarajan [2] , James C. Storey [2] and Mark Lubke [2]**

1    U.S. Geological Survey, Earth Resources Observation and Science Center, Sioux Falls, SD 57030, USA
2    KBR, Contractor to the U.S. Geological Survey, Earth Resources Observation and Science Center, Sioux Falls, SD 57030, USA
*    Correspondence: choate@usgs.gov

**Abstract:** The Landsat program has a long history of providing remotely sensed data to the user community. This history is being extended with the addition of the Landsat 9 satellite, which closely mimics the Landsat 8 satellite and its instruments. These satellites contain two instruments, the Operational Land Imager (OLI) and the Thermal Infrared Sensor (TIRS). OLI is a push-broom sensor that collects visible and near-infrared (VNIR) and short-wave infrared (SWIR) wavelengths at 30 m ground sample distance, along with a panchromatic 15 m band. The TIRS sensor contains two long-wave thermal spectral channels centered at 10.9 and 12 μm. The data from these two instruments, on both satellites, are combined into a single Landsat product. The Landsat 5–9 satellites follow a 16 day repeat cycle designated as the Worldwide Reference System (WRS-2), which provides a global notional gridded mapping for identifying individual Landsat scenes. The Landsat 8 and 9 satellites are flown such that their orbital tracks are separated by 8 days in this 16 day cycle. During the commissioning period of Landsat 9, and during its ascent to its operational WRS-2 orbit, the Landsat 9 satellite's orbital track went under and crossed over the orbital track of the Landsat 8 satellite. This produced a unique situation where nearly time-coincident imagery could be obtained from the instruments of the two spacecrafts. From a radiometric standpoint, this allowed for near-time cross-calibration between the instruments to be performed. From a geometry perspective, calibration is achieved through high-resolution reference imagery over specific ground locations, thus ensuring calibration of the instruments and for the instruments to be well cross-calibrated geometrically. Although these underfly data do not provide calibration of the instruments between the platforms from a geometric perspective, they allow for the verification of the calibration steps involving the instruments and spacecraft. This paper discusses the co-registration of this unique set of data while also discussing other geometric aspects of these data by looking at and comparing the differences in sensor viewing and sun angles associated with the collections from the two platforms for imagery obtained over common geographic locations. The image-to-image comparisons between Landsat 8 and 9 coincident pairs, where both datasets are precision terrain products, are registered to within 2.2 m with respect to their root-mean-squared radial error (RMSEr). The 2.2 m represents less than 0.1 of a 30 m multispectral pixel in misregistration between the L9 and L8 underfly products that will be available to the user community. This unique dataset will provide well-registered, near-coincident image acquisitions between the two platforms that can be a key to any calibration or application comparisons. The paper also presents that, for images for which one of the image pairs failed precision corrections and became a terrain-corrected only product type, a range of 8–14 m RMSEr could be expected in co-registration, while, in cases where both image pairs failed the precision correction step and both images became a terrain-corrected only product type, a 14 m RMSEr could be expected for co-registration.

**Keywords:** Landsat 8; Landsat 9; Operational Land Imager; Thermal Infrared Sensor; image co-registration; satellite viewing geometry

## 1. Introduction

The Landsat program has a long heritage of acquiring, archiving, and distributing remotely sensed data [1]. Starting in 2016, the US Geological Survey (USGS) adopted a tiered collection management structure for its Landsat data products that ensures a consistent set of processing for the Landsat archive within a given collection, while allowing a set of calibration updates to be performed between any two given collections [2]. This collection philosophy provides the user community with a set of Level-1 and Level-2 products for the entire Landsat archive that is consistent not only in product format, but also in the radiometric and geometric algorithms used in their processing. A key part of integrating the Landsat 9 satellite and its sensors into this structure, from a radiometry perspective, involves the datasets collected during the Landsat 9 (L9) underfly of Landsat 8 (L8). The nearly time-coincident acquisitions of the imagery help alleviate time-varying components, such as atmospheric and temporal changes, that impact the ability to otherwise compare products from the two satellites more directly from a radiometry perspective. Similarly, this coincident imagery also provides better co-registration between the products generated from the two systems by eliminating the impacts of those same time-varying components from the differences that may arise during registration and measuring the systematic corrected imagery to the ground control. Once these temporal differences and changes are reduced, other aspects of the differences between the systems such as sun and sensor viewing angles can become more pronounced in the direct comparisons between the two systems. This paper describes some of the aspects of the underfly from an acquisition perspective, the co-registration between the Level-1 products produced from the two systems, and the driving forces in the differences between sensor viewing and sun angles associated with the products, giving some examples of these differences. Within this paper, these differences are only addressed for the multispectral data of the Operational Land Imager (OLI) instruments.

### 1.1. Operation Land Imager Architecture

The Operational Land Imager is made up of 14 sensor chip assemblies (SCAs), staggered in the along-track direction, where each band is aligned with respect to the even and odd SCAs in the across-track direction. This includes minor clocking of the SCAs to compensate for optical distortion. Figure 1 shows the SCA staggering and band ordering within an SCA of the OLI instrument [3].

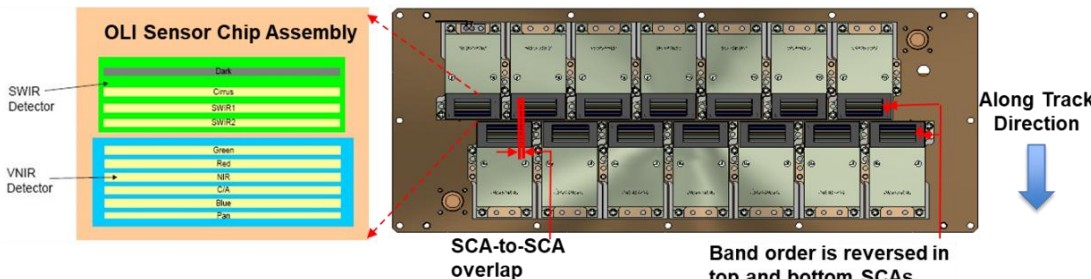

**Figure 1.** Operational Land Imager (OLI) focal plane layout. Each sensor chip assembly (SCA) contains linear arrays, the combination of which, in the across-track direction, determines the across-track field of view. Bands are staggered in the along-track direction within a given SCA. The OLI sensor chip assembly includes short-wave infrared (SWIR) and visible and near-infrared (VNIR) detectors.

This staggering in the along-track direction produces a noncontiguous geographic sampling across track within the odd and even SCA groupings, while remaining contiguous within an SCA across the 494 detectors in each band of a SCA. This along-track staggering can be represented as the number of nominal multispectral lines between the trailing and leading SCAs or the odd and even numbered SCAs, for each band in the along-track direction, and the corresponding timing delay when aligning the SCAs geographically. Figure 2 shows this SCA staggering within an image, caused by the focal plane displacement

of each band within each SCA. For Figure 2, by not trimming the leading and trailing portions of the imagery acquired at the same time from the SCAs across bands during a given image acquisition in the output product, while also listing timing differences between the leading and trailing SCAs, the geographic displacement for a single frame of imagery acquired can be shown along with the timing differences between when the leading and trailing bands within an SCA acquire data within a collection. The timing in the table, therefore, represents the maximum relative timing delay of the odd and even SCAs for a given band. An item that is worth noting from Figure 1 is that the staggering and placement of the SCAs indicate that the nominal nadir pointing of the instrument does not truly lie within any given SCA but lies within the grouping of the SCAs placed within the center of the focal plane.

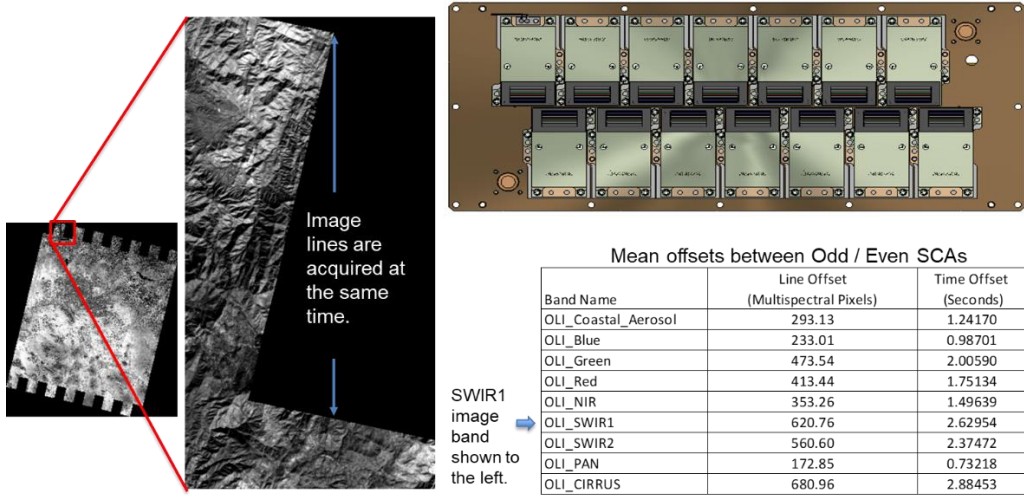

Mean offsets between Odd / Even SCAs

| Band Name | Line Offset (Multispectral Pixels) | Time Offset (Seconds) |
|---|---|---|
| OLI_Coastal_Aerosol | 293.13 | 1.24170 |
| OLI_Blue | 233.01 | 0.98701 |
| OLI_Green | 473.54 | 2.00590 |
| OLI_Red | 413.44 | 1.75134 |
| OLI_NIR | 353.26 | 1.49639 |
| OLI_SWIR1 | 620.76 | 2.62954 |
| OLI_SWIR2 | 560.60 | 2.37472 |
| OLI_PAN | 172.85 | 0.73218 |
| OLI_CIRRUS | 680.96 | 2.88453 |

**Figure 2.** Geometrically corrected image of band 6 of the Operational Land Imager demonstrating geographically the along-track staggering between the odd and even sensor chip assemblies (SCAs). A table showing the mean offsets between the odd and even SCAs in multispectral lines and seconds is shown to the right of the image.

This time delay between the sampling of the even and odd SCAs means that the sun angles and sensor angles will produce discontinuous values across track within the imagery. For sun angles, these timing differences produce small differences. Even though the timing differences between two acquisitions for the L9 and L8 underfly imagery are larger than those within scenes due to the staggering of the SCAs, the sun angle differences are still considered small between the two acquisitions. For the sensor viewing angles, especially for the azimuth viewing angles, which are dependent on the focal plane SCA placement over the geographic area acquired, discontinuities within an image can be large, as demonstrated in the analysis results presented within this paper [4].

*1.2. Sensor and Sun Angle Viewing File*

Within this paper, a Level-1R (L1R) refers to a dataset that has been radiometrically corrected but does not have any geometric corrections applied to the imagery. Level-1T (L1T) refers to imagery that is radiometrically and geometrically corrected and has corrections applied for the terrain parallax effects within the imagery using a digital elevation model (DEM). This L1T referenced within this paper can be either a Level-1 Terrain and Precision (L1TP) or a Level-1 Systematic Terrain (L1GT) image. The L1TP image has terrain effects removed from the imagery and has biases present in the spacecraft telemetry data, position, and attitude, removed on the basis of measurements made between the imagery and the Landsat Global Survey (LGS) ground control library. The L1GT has the terrain effects removed from the imagery but the data do not have the spacecraft biases associated with its telemetry data removed. When speaking to Landsat 9 and 8 products within this paper, many can be part of both the L1TP and the L1GT products; where it is

unnecessary to designate between the two product types, L1T is used, whereas, in cases where it is necessary to differentiate, and a subject can only be referred to with either the L1TP or the L1GT nomenclature, that designation is used.

Each Landsat product is accompanied by a solar illumination and sensor viewing angle coefficient file that contains sensor viewing and sun angle information for that product. The procedure for using the file contents to compute the angular relationships is performed through the following steps: (1) calculating an image time (per pixel) by determining a Level-1R line number, which can be directly related to time, using a rational polynomial relating Level-1T product coordinates to the corresponding Level-1R line number; (2) evaluating an ephemeris model that provides spacecraft position in an Earth-centered Earth-fixed (ECEF) coordinate system using the Level-1R line number as a proxy for time as an input; (3) evaluating a sun direction model that provides solar coordinates as an ECEF unit vector given a Level-1R line/time.

The rational polynomial relationship between the Level-1T and Level-1R image coordinates is shown in Equation (1). The height in the rational polynomial equations is the height at the given Landsat product pixel location, extracted from a DEM. The equations listed below, Equations (1) and (2), are determined from the spacecraft and instrument telemetry along with the spacecraft and instrument characteristics such as the instrument focal plane design. The USGS provides the user community the tools, using the equations listed in this paper, that are needed to generate their own set of a sensor viewing and sun angles for Level-1 products. [4]. For the equations listed below, the nomenclature of L1T refers the Level-1T image product, either L1GT or L1TP. The subscripts Line and Sample refer to the Level-1T product line and sample pixel locations within the output product. The MeanLine and MeanSample subscripts are the mean line and sample pixel locations within the L1T product. The rational polynomial coefficients given in Equation (1), $a_n$, $b_n$, $c_n$, and $d_n$, allow the user to calculate time by determining the L1R pixel location of a given L1T product line and sample pixel location. The determination of the L1R pixel location allows the rational polynomial coefficients $e_n$ and $f_n$ to be used to determine the satellite's position or the sun's position for a given L1T output pixel location. This produces a flow of first determining the L1R pixel location and time for a given L1T output pixel location. Once time is determined, the sun's location and the satellite's position can be found for a given L1T output pixel location. This information can then be used to determine the corresponding viewing and sun angles for that L1T output pixel location. The satellite and sun's position calculated resides in an Earth-centered Earth-fixed coordinate system with units of meters.

$$
\begin{aligned}
L1R_{Line} &= L1R_{MeanLine} + \frac{a_0 + a_1 * L1T_L + a_2 * L1T_S + a_3 * Hgt + a_4 * L1T_L * L1T_S}{1 + b_1 * L1T_L + b_2 * L1T_S + b_3 * Hgt + b_4 * L1T_L * L1T_S}, \\
L1R_{Sample} &= L1R_{MeanSample} + \frac{c_0 + c_1 * L1T_L + c_2 * L1T_S + c_3 * Hgt + c_4 * L1T_L * L1T_S}{1 + d_1 * L1T_L + d_2 * L1T_S + d_3 * Hgt + d_4 * L1T_L * L1T_S},
\end{aligned}
\tag{1}
$$

where $L1T_L = L1T_{Line} - L1T_{MeanLine}$, $L1T_S = L1T_{Sample} - L1T_{MeanSample}$, $L1T_{Line}$ is the output pixel line location in $L1T$, $L1T_{Sample}$ is the output pixel sample location in $L1T$, $Hgt = Height - Height_{Mean}$, $L1R$ is the Level-1R pixel location, $L1T$ is the Level-1T pixel location, $Height$ is the elevation at the $L1T$ pixel location, $L1T_{MeanSample}$ is the mean sample pixel location in $L1T$, $L1T_{MeanLine}$ is the mean line pixel location in $L1T$, $Height_{Mean}$ is the mean elevation in $L1T$, and $a_0$ to $a_4$, $b_1$ to $b_4$, $c_0$ to $c_4$, and $d_1$ to $d_4$ are rational polynomial coefficients (RPCs) of the model.

Equation (1) provides rational polynomials defining Level-1 terrain corrected image (L1T) line and sample locations to input Level-1R geometrically raw (uncorrected) line and sample locations. By determining the Level-1R geometrically uncorrected pixel location, the time for a given L1T line and sample location can be calculated.

The equation defining the satellite position on the basis of the Level-1T and Level-1R image, with the Level-1R location determined by Equation (1), is shown in Equation (2). Equations for the other satellite coordinate axis locations ($Sat_y$ and $Sat_z$) and the sun's

corresponding cartesian coordinates ($Sun_x$, $Sun_y$, and $Sun_z$) follow the same format as below for the satellite's X ($Sat_x$) location.

$$
\begin{aligned}
Sat_x &= Sat_{XMean} \\
&+ \frac{e_0 + e_1*L1T_L + e_2*L1T_S + e_3*Hgt + e_4*L1R_L + e_5*L1T_L^2 + e_6*L1T_L + e_7*L1T_s^2 + e_8*L1R_S*L1R_L^2 + e_9*L1R_L^3}{1 + f_1*L1T_L + f_2*L1T_S + f_3*Hgt + f_4*L1R_L + f_5*L1T_L^2 + f_6*L1T_L + f_7*L1T_s^2 + f_8*L1R_S*L1R_L^2 + f_9*L1R_L^3}
\end{aligned}
\tag{2}
$$

where $Sat_X$ is the satellite Earth-centered Earth-fixed *X*-coordinate, $Sat_{XMean}$ is the satellite Earth-centered Earth-fixed mean *X*-coordinate, $L1T_L = L1T_{Line} - L1T_{MeanLine}$, $L1T_S = L1T_{Sample} - L1T_{MeanSample}$, $L1T_{Line}$ is the output pixel line location in $L1T$, $L1T_{Sample}$ is the output pixel sample location in $L1T$, $Hgt = Height - Height_{Mean}$, $L1R_L = L1R_{Line} - L1R_{MeanLine}$, $L1R_S = L1R_{Sample} - L1R_{MeanSample}$, $L1R$ is the Level-1R pixel location, $L1T$ is the Level-1T pixel location, *Height* is the elevation at output L1T pixel location, $L1T_{MeanSample}$ is the mean sample pixel location in $L1T$, $L1T_{MeanLine}$ is the mean line pixel location in $L1T$, $Height_{Mean}$ is the mean elevation in $L1T$, and $e_0$ to $e_9$ and $f_1$ to $f_9$ are RPCs of the model.

Equation (2) provides rational polynomials defining satellite location given a Level-1R, an image that is radiometrically corrected but geometrically uncorrected, and a Level-1T, an image that has terrain compensation and then may or may not contain corrections based on the application of ground control, t line, and sample location. These relationships define the satellite *X*-coordinate position. The equations for the satellite *Y*- and *Z*-coordinate positions follow the same format, with a different set of rational polynomial coefficients. The satellite's coordinate location at a given time can in turn define the viewing angle for a given L1T output pixel location.

*1.3. Landsat 9 Underfly*

The Landsat 9 underfly refers to the period during the commissioning of the Landsat 9 satellite and its instruments that allowed L9 to slowly drift under the orbital track of the Landsat 8 satellite. This was observed during the commissioning period when a series of orbital maneuvers (termed "burns") were needed to raise the orbit of the satellite to its final destination within the Worldwide Reference System 2 (WRS-2). [5]. The exact timeframe during which the L9 satellite's sensors overlapped with those of its L8 counterpart depends upon what is used to define an adequate amount of overlap between the systems for a given application. Typically, the defined time-frame for the underfly resides across 5 days where there was at least a 10% overlap between the fields of view of the instruments on the two satellites. However, since the amount of overlap also varies with latitude, even this 5 day window can be somewhat misleading, as represented in Figure 3.

In Figure 3, it is demonstrated that, at higher latitudes, there is more overlap between the field of view of the two satellites and its sensors than at mid and lower latitudes. At the beginning and end of the underfly period, overlap may only be present at high-latitude locations. To demonstrate the amount of overlap and number of scenes acquired during the underfly period Figure 4 was assembled to show the percentage overlap with respect to the geographic location of the acquisitions. Blue in Figure 4 represents the scenes with a limited amount of overlap, while red represents scenes with the maximum amount of overlap present in the image pairs.

Along with the percentage overlap between the two acquisitions, another aspect that can be important in data analysis is that, at the beginning of the underfly, as well as at the end of the underfly, the time difference between the two acquisitions was greater than at the heart of the underfly, i.e., the time in the middle of the underfly date range. These time differences lead to small differences in the sun angles associated with the acquisitions. While those differences are not in the range of multiple days as would be the case in normal operations, but rather minutes, this change will nevertheless produce differences in the sun angles associated with the two acquisitions. The time differences between scenes relative to the L8 acquisition time are shown in Figure 5 across the date range of the underfly. These time differences were based on times extracted from metadata information listing scene

center times to the nearest whole second. The plot in Figure 5 shows the difference in scene center times over the range of date of the underfly. Jumps and discontinuities in the plot are associated with the fidelity of the scene times being kept. Taking into account this granularity in measurements, the time difference between L9 and L8 changed by approximately 0.002209 s/s or 190.8623 s/day.

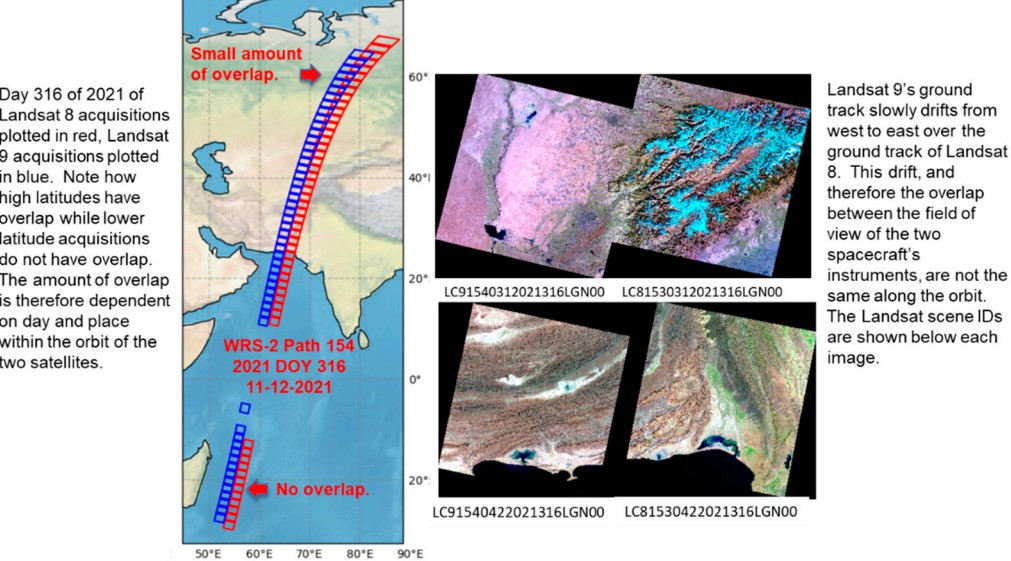

**Figure 3.** Scene corners for one pass during the early time-frame of the Landsat 9 underfly. Red boxes are the bounds of Landsat 8 acquisitions. Blue boxes are the bounds of Landsat 9 acquisitions. High-latitude scenes contain partial overlap, while mid- to lower-latitude scenes do not contain any overlap.

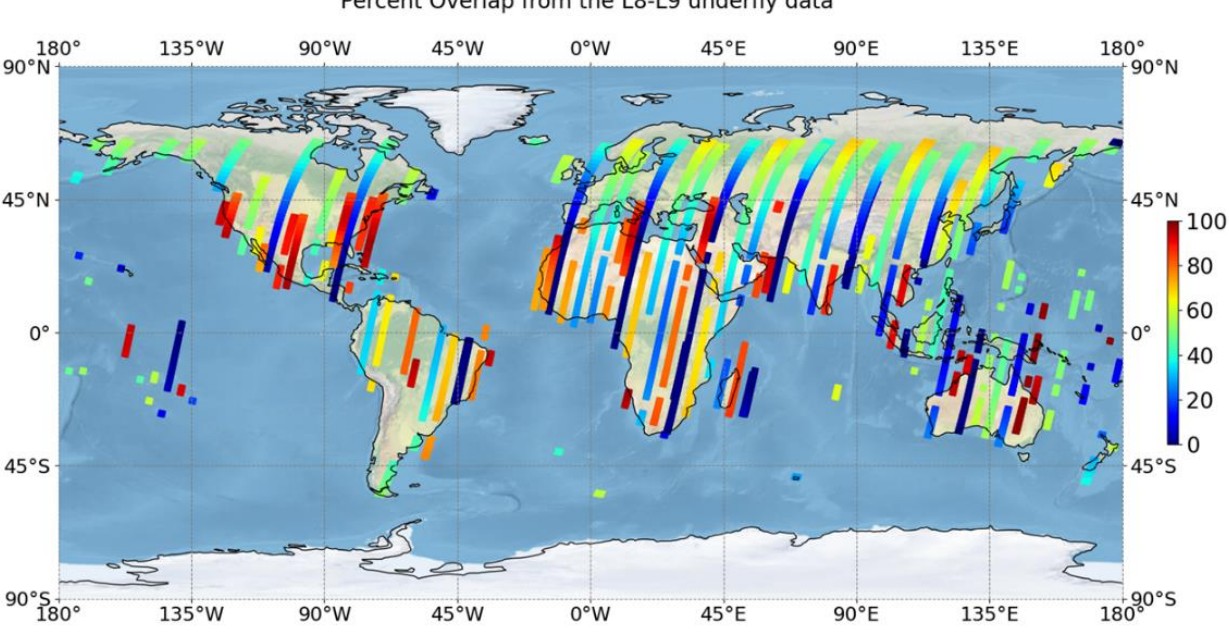

**Figure 4.** Percentage overlap between Landsat 9 and Landsat 8 acquisitions during the underfly period based on dates 12 November 2021 to 17 November 2021. Percentages are based on geographic areas of the two acquisitions and that are within −10 to +8 min in acquisition times. Only descending passes are shown.

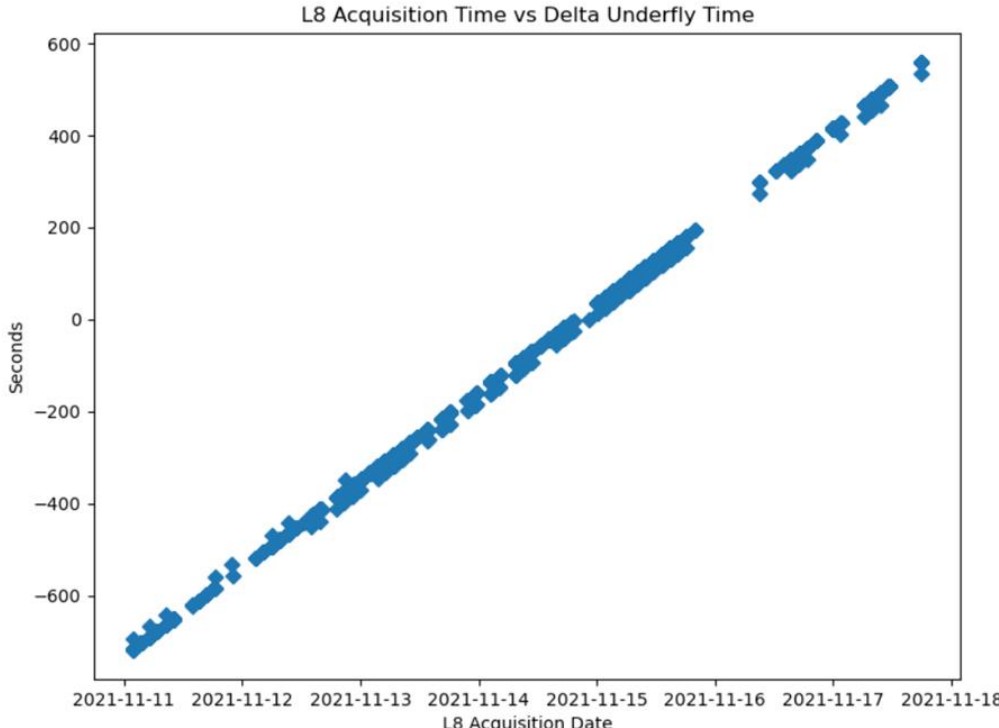

**Figure 5.** Time differences between Landsat 9 and Landsat 8 scene center times determined from the product metadata information. These scene center times were determined to a whole second resolution. The *X*-axis is the L8 scene center time.

## 2. Materials and Method

The USGS Landsat Image Assessment System (IAS) performs calibration, characterization, and validation of Landsat satellites, sensors, and products, across all missions and all sensors [6–9]. One of the functions of this system is to register the systematically corrected imagery to the Landsat Ground Control [10]. This ground control is used across all missions and provides a globally distributed and consistent set of control. The IAS provides feedback on the results of this registration while also having the ability to perform a direct image-to-image (I2I) registration accuracy characterization between image products using a correlation-based mensuration process. The IAS I2I registration characterization is performed by extracting windows of imagery from both the L8 and the L9 data sets at common geographically co-located areas. A normalized grayscale correlation is performed over these windowed areas, outliers are removed on the basis of the correlation peak strength, and a Student *t*-test is performed on the correlation offsets calculated after the inspection of the correlation peak to further remove outliers. The remaining measured correlations are used to calculate the statistics shown in the tables and results within this paper. This I2I characterization process was used to determine the co-registration properties of products produced from the underfly data. As the geometric algorithms and steps used in the IAS, along with the use of the Landsat Ground Control, are also applied in the Landsat Product Generation System (LPGS), the results from the analysis in this paper also represent the co-registration that can be expected for the publicly distributed data over this underfly time-frame. For the I2I analysis performed within this paper, initial scene selection was based solely on cloud screening and the amount of overlap between the image pairs. Once the results from the I2I steps were determined, scenes with poor correlation features, such as a lack of content or texture within the imagery, were removed as outliers due to their inability to produce good correlation results based on scene content.

The LPGS creates Level-1 and Level-2 Landsat products for the user community. The products generated from this system contain the solar illumination and sensor viewing angle coefficient files (ANG) [4]. As noted above, this text file contains information, rep-

resented in rational polynomials, that allows the user, through a USGS distributed tool, to generate image files that contain per-pixel sun azimuth and zenith viewing angles along with sensor azimuth and zenith viewing angles. These equations were discussed in Section 1.2.

Generating these angles for the OLI instrument and understanding their characteristics are particularly challenging due to the OLI sensor's multiple-SCA architecture. As discussed previously, the instrument obtains a full field of view through multiple staggered sensor arrays or SCAs. These SCAs are staggered in both the along- and the across-track directions. Bands are offset in the along-track direction creating parallax differences between each SCA and band, while an across-track staggering is used to obtain the full field of view of the products generated. This causes noncontiguous sampling of the data in time once the SCAs are stitched together geographically to produce a continuous across-track product geometrically. The USGS provides a standalone tool that reads the angle metadata files associated with the sun and sensor angle geometry and allows the user to create output images that contain the azimuth and zenith sensor viewing geometry, as well as the azimuth and zenith sun angles in degrees scaled by a factor of 1/100. The images created from this tool are map-projected and co-registered with the Level-1 multispectral image products that they are associated with [4].

## 3. Results and Discussion

As discussed previously, the co-registration accuracy between the L9 and L8 imagery during the underfly is based on their common registration to the Landsat Ground Control Library GCPs. Updates to these GCPs throughout the collection phased releases of Landsat products and software have increased the accuracy of this reference dataset where an accuracy of better than 9.5 m (CE95) can be expected in areas of limited cloud cover and temporal changes between the search and reference imagery [11–14]. The co-registration between the L9 (search) and L8 (reference) image products should have essentially no differences due to temporal changes between the search and reference imagery, and any differences that may be present in the co-registration would be due to the regions in the imagery that are not part of the overlap between the fields of view, where ground control will be applied to a geographic area for one scene and not the other. These differences in the two scenes of ground coverage between the reference and search imagery lead to differences in the set of control points used and may impact the precision solution step of correcting for any systematic biases present in the spacecraft telemetry data. However, these differences between the two datasets produced in the precision step are expected to be small. Table 1 shows the image-to-image (I2I) statistics generated on the IAS for each band of data with at least 50% land within the imagery, less than 20% cloud cover within the imagery, and at least 10% overlap between the L9 and L8 products. Statistics are listed in units of meters. A total of at least 388 scenes were used in the analysis. As individual band results were filtered for outliers, and as different bands will have differing image-to-image correlation results due to the spectral band differences and the land features acquired during imaging, the number of images used in calculating the results is not the same across all bands. The table shows the mean, standard deviation (STD), and root-mean-squared error (RMSE) given in units meters. From Table 1, the imagery is co-registered to less than 2.2 m radial root-mean-squared error (RMSEr) for all bands.

Figure 6 shows the geographic distribution of the scenes from Table 1. As shown in the figure, the scene list was well distributed geographically.

To investigate whether there is a dependency between the co-registration accuracy and the amount of overlap between the two images, the RMSEr based on the percentage overlap was analyzed, as shown in Figure 7. Each image pair's radial RMSE from I2I is shown along the *y*-axis, while the percentage overlap between the images is shown along the *x*-axis.

**Table 1.** Image-to-image registration statistics for 388+ Landsat 9 and Landsat 8 images acquired during the underfly. Images used in analysis contained at least 10% overlap between the Landsat 8 and 9 images and were Level-1 Terrain Precision (L1TP) products. Mean, standard deviation (STD), root-mean-squared error (RMSE), and radial RMSE (RMSEr) in units of meters are shown.

| Image-to-Image (Meters) | | | | | | | | | | |
|---|---|---|---|---|---|---|---|---|---|---|
| Band | Num Scenes | Mean Line | Mean Sample | STD of Mean Line | STD of Mean Sample | STD Line | STD Sample | RMSE Line | RMSE Sample | RMSEr |
| 1 | 394 | −0.36 | 0.28 | 1.29 | 0.85 | 1.02 | 0.84 | 1.69 | 1.23 | 2.09 |
| 2 | 397 | −0.28 | 0.29 | 1.40 | 0.85 | 1.09 | 0.83 | 1.79 | 1.22 | 2.17 |
| 3 | 397 | −0.18 | 0.27 | 1.32 | 0.89 | 1.06 | 0.84 | 1.70 | 1.25 | 2.11 |
| 4 | 398 | −0.05 | 0.29 | 1.31 | 0.89 | 1.05 | 0.82 | 1.68 | 1.24 | 2.09 |
| 5 | 396 | −0.07 | 0.21 | 1.31 | 0.92 | 1.04 | 0.82 | 1.67 | 1.25 | 2.09 |
| 6 | 395 | 0.04 | 0.22 | 1.38 | 0.93 | 1.06 | 0.86 | 1.74 | 1.29 | 2.16 |
| 7 | 397 | −0.02 | 0.25 | 1.31 | 0.93 | 1.05 | 0.87 | 1.67 | 1.29 | 2.12 |
| 8 | 388 | −0.20 | 0.25 | 1.27 | 0.75 | 0.78 | 0.73 | 1.50 | 1.07 | 1.85 |

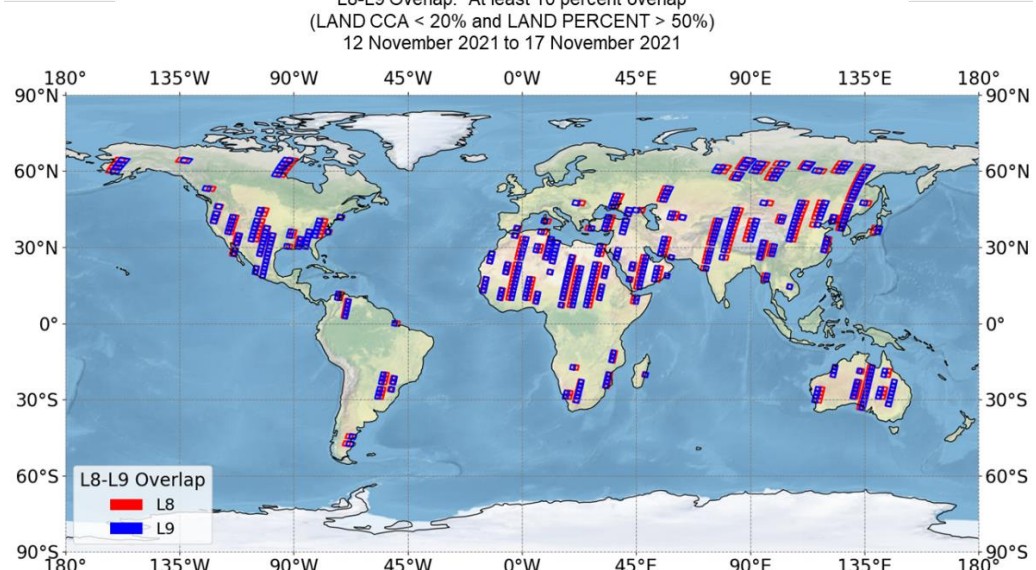

**Figure 6.** Scenes from the Landsat 8 and 9 underfly time-frame with at least 10% overlap, at least 50% land, and less than 20% cloud cover. These image pairs were used in an image-to-image registration assessment of the two Landsat products during the underfly period.

Figure 7 shows that the amount of overlap between the L8 and L9 images had essentially no impact on the co-registration of the imagery within the overlap region. This assessment shows that, regardless of the overlap within the two products, the registration between the two products will essentially be the same due to the use of a common, consistent control point reference.

Another indicator of the registration accuracy of the Landsat products is the results generated from the geodetic accuracy assessment. This step determines how well the imagery fits to the Landsat ground control both before (pre-fit) and after (post-fit) adjusting for any bias or rate errors present in the spacecraft's telemetry for position or pointing knowledge. Although this indicator represents the relationship between the imagery and the ground control, rather than the previous I2I results, which demonstrates how well the imagery stacks together from a co-registration perspective, the geodetic accuracy results provide useful information to support the I2I results and can indicate the geometric accuracy of an L1GT image product, an image that has the effects of terrain removed but no ground control applied to the data. Using the Landsat 9 and 8 I2I scene list and determining the overall geodetic accuracy pre-fit and post-fit results along and across track for those scenes produced the results listed in Table 2. The RMSE listed in the table is the RMS of the RMSE of all the scenes analyzed.

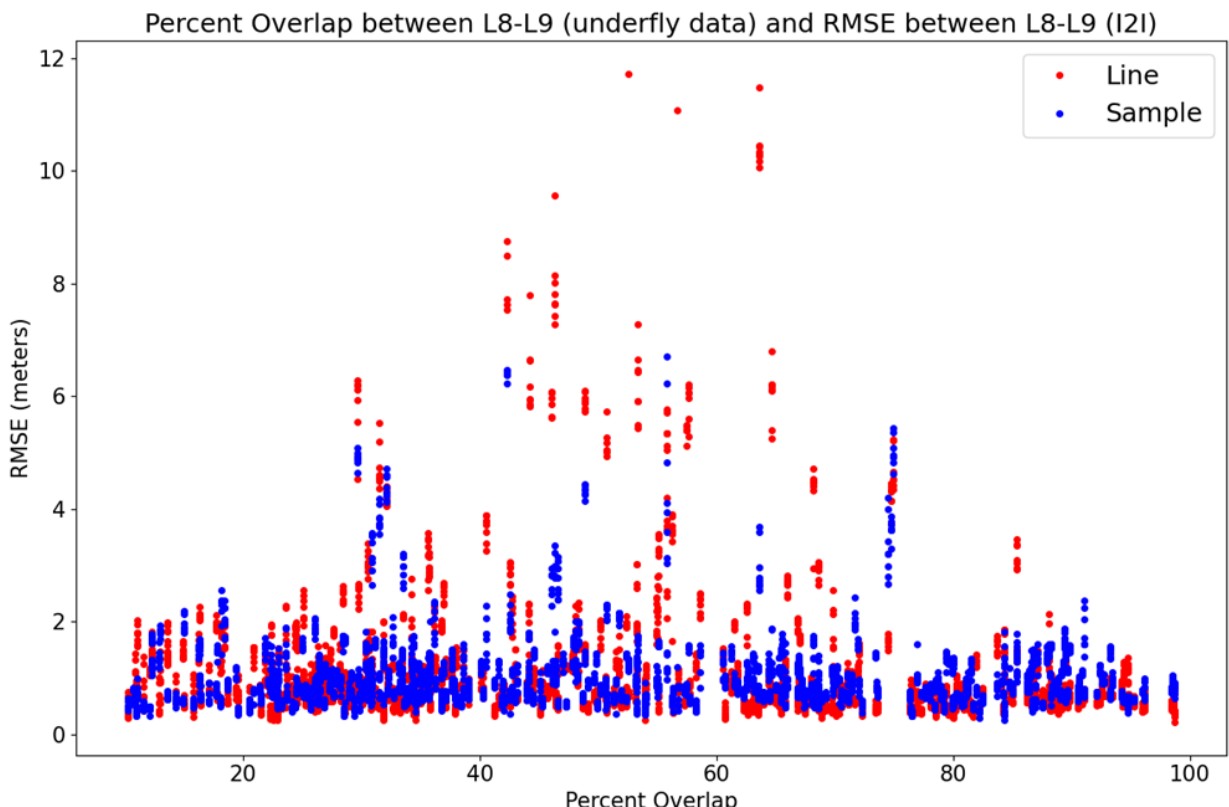

**Figure 7.** Radial root-mean-squared error (RMSEr) measured using image-to-image correlation between Landsat 8 (L8) and 9 (L9) image products from the underfly time-frame. Image pairs with at least 10% overlap, less than 20% land cloud cover, and at least 50% land in the scene were used in the analysis. The graph shows the relationship between the RMSE (radial) and the percent overlap between the L8 and L9 scenes.

Table 2 shows that both sets of instruments (L9 and L8 OLI instruments) were very consistent with the Landsat Ground Control library. This is based on the low RMSEs for both the pre- and post-fit results. The pre-fit results also help indicate what could be expected for any imagery from the underfly time-frame which may not be able to achieve a product level of precision correction and is available in an L1GT state. A question regarding L1GT product registration accuracy, i.e., for scenes that do not go to an L1TP product, but fall back to an L1GT, is how well they will be registered to each other. Scenes that are distributed as L1GT products that either are in areas where no ground control points are available (e.g., Antarctica) or fail to reach a precision state due to a lack of good clear ground cover because of clouds, snow, or landscape changes with respect to the ground control used and the newly acquired data. Landsat 8 has shown very good registration accuracy without the ability to apply ground control, typically within the 18–22 m CE90 range [15]. During the early commissioning of L9, the same time-frame as the underfly period, L9 tended to exhibit similar characteristics. To determine the co-registration of the L1GT underfly scenes, an analysis was performed by processing a group of L1TPs, 40 cloud-free scenes, to an L1GT state, and an I2I analysis was performed on these scenes produced with the two differing product types, comparing L1TPs to L1GTs. These 40 scenes, now in an L1GT state, were compared using the same I2I processes used to compare the L1TPs, i.e., the previous I2I analysis performed and shown in Table 1. The L9 L1GTs were compared against the L8 L1TPs by performing I2I between these 40 image pairs. Those results were then compared against the pre-fit geodetic accuracy reports generated from the L9 L1TPs. This comparison was conducted to verify that the pre-fit geodetic results provided from the L9 L1TPs would be an indication of what the co-registration accuracy of an L9 L1GT would be to the L8 L1TPs. As the pre-fit geodetic accuracy results are a

measure of the geometric offsets between the precision-corrected L1GT and the Landsat ground control, measuring the L9 L1GTs to the L8 L1TPs that were also registered to the same ground control should essentially give the same results. A plot of this relationship is shown in Figure 8, where the L9 pre-fit geodetic accuracy radial offset (RMSEr) results are shown along the *y*-axis, while the radial offset of the I2I results are shown along the *x*-axis. As the reports for the geodetic accuracy are given in terms of the along- and across-track direction and the I2I results are in map projection, line and sample coordinates looking at the radial components will help alleviate differences that would arise from plotting along and across track with the map-projected *X* and *Y* offsets. These results should show good correlation between the measurements, as the radial offset in an ideal situation between the two measurements should be equivalent since we expect the geometric differences between any given L1GT and the Landsat ground control library to be equivalent to the difference between that L1GT and an L1TP over the same area. Differences in the sampling of the measurements made within the two scenes between the two processes, with differing outlier logic performed on the measurements, would introduce some differences between the I2I and pre-fit geodetic accuracy results. As can be seen in Figure 8, the correlation between the pre-fit geodetic accuracy and I2I results for all bands was 0.93. This figure demonstrates that L9 scenes that may not achieve an L1TP state and fall back to L1GT are well represented by the pre-fit geodetic accuracy results shown in Table 2. Along with the outlier logic differences stated previously, the I2I results would produce a better measurement in that the I2I characterization approaches an almost auto correlation between two datasets, with only small differences in radiometry and geometry being present between the imagery. Conversely, the geodetic accuracy involves the correlation between the ground control library and the L9 imagery acquired; this correlation step will also include any temporal and landscape changes. Although these temporal type differences should be minimized by the geodetic accuracy step filtering poor correlations based on the ability to produce a precision model, it is the I2I correlation step that will provide a purer measurement from a correlation perspective between the images.

**Table 2.** Along- and across-track geodetic accuracy for Landsat 9 and 8 scenes used in image-to-image analysis study. Pre-fit results are the registration accuracy prior to applying corrections to the imagery based on ground control, while post-fit results are after applying the ground control to correct for any biases or rate errors present in the spacecraft telemetry. Values shown are in units of meters. Results are generated on the basis of the precision correction of the Level-1 Systematic Terrain products (L1GT).

| Geodetic Accuracy Statistics | | | | | | | |
|---|---|---|---|---|---|---|---|
| | | Mean of Means (Meters) | | Standard Deviations of Means (Meters) | | RMSE (Meters) | |
| Satellite | Type | Along Track | Across Track | Along Track | Across Track | Along Track | Across Track |
| L9 | Post-Fit | 0.0001 | 0.0011 | 0.0028 | 0.0038 | 5.1984 | 4.7307 |
| L8 | Post-Fit | 0.0010 | −0.0014 | 0.0113 | 0.0040 | 5.2605 | 4.7427 |
| L9 | Pre-Fit | 2.1588 | 3.6276 | 8.8641 | 4.3506 | 7.5697 | 7.5524 |
| L8 | Pre-Fit | 4.2831 | −6.0087 | 12.6848 | 6.6966 | 14.6342 | 10.2913 |

The second analysis involving the L1GTs was to compare the L9 and L8 L1GTs themselves. The same 40 scenes used in the L9 L1GT to L8 L1TP comparison were used in the matching L1GT comparison. Table 3 shows the I2I results between the L9 and L8 L1GTs.

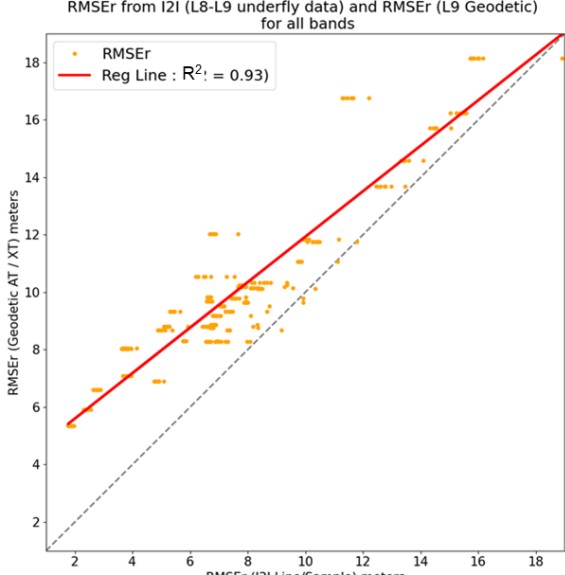

**Figure 8.** Comparison between the pre-fit geodetic accuracy results for 40 scenes during the L9 underfly period and image-to-image comparisons between the Landsat 8 Level-1 Precision and Terrain (L1TP) and L9 Level-1 Systematic Terrain images (L1GT). This plot shows good correlation between the two measurements indicating, that the pre-fit geodetic accuracy results show the expected registration accuracy for L9 scenes that may not reach an L1TP state and fall back to an L1GT.

**Table 3.** Image-to-image measured offsets between 40 Landsat 9 and Landsat 8 Level-1 Systematic Terrain Images (L1GT). These images were acquired during the Landsat 9 underfly of Landsat 8 period.

| | | | | Image-to-Image (Meters) | | | | | | |
|---|---|---|---|---|---|---|---|---|---|---|
| Band | Num Scenes | Mean Line | Mean Sample | STD of Mean Line | STD of Mean Sample | STD Line | STD Sample | RMSE Line | RMSE Sample | RMSEr |
| 1 | 40 | −2.81 | −11.21 | 5.15 | 7.22 | 5.15 | 7.22 | 5.98 | 13.36 | 14.64 |
| 2 | 40 | −2.88 | −11.21 | 5.15 | 7.22 | 5.15 | 7.22 | 6.01 | 13.36 | 14.65 |
| 3 | 40 | −2.98 | −11.16 | 5.17 | 7.23 | 5.17 | 7.23 | 6.06 | 13.32 | 14.63 |
| 4 | 40 | −3.13 | −11.20 | 5.19 | 7.22 | 5.19 | 7.22 | 6.15 | 13.34 | 14.69 |
| 5 | 40 | −3.10 | −11.07 | 5.16 | 7.14 | 5.16 | 7.14 | 6.11 | 13.19 | 14.54 |
| 6 | 40 | −3.18 | −11.09 | 5.20 | 7.13 | 5.20 | 7.13 | 6.20 | 13.21 | 14.59 |
| 7 | 40 | −3.14 | −11.20 | 5.21 | 7.16 | 5.21 | 7.16 | 6.18 | 13.31 | 14.67 |
| 8 | 40 | −3.22 | −11.36 | 5.53 | 6.52 | 5.53 | 6.52 | 6.46 | 13.10 | 14.61 |

With the co-registration being a little more than 2 m for bands in the underfly image L1TP pairs, making the underfly data well co-registered, one element that could play a prominent role in comparisons of the imaging pairs from a science perspective is the sun and sensor viewing angle differences. As stated previously, the sensor architecture of the Landsat 9 and 8 instruments makes the sun and sensor viewing angles for the images acquired complex. Due to the focal plane of the OLI instrument being made up of 14 independent linear arrays that together form the full across-track field of view, the sun and sensor viewing angles are not contiguous in time of data acquisition when one traverses from east to west, or vice versa, across an image product. Figure 9 shows the L9 and L8 underfly scenes from WRS-2 037/037 acquired on 15 November 2021. The overlap between these two scenes was 64%, while the acquisition time difference between the scene center times was 179.759 s. The L9 sensor azimuth angles shown in the upper right of the figure demonstrate the SCA-to-SCA discontinuity between the set of linear arrays that make up the focal plane for a given band. This discontinuity, which is most visible in the figure for the sensor azimuth angles, is present in all angle bands generated. This

discontinuity in angles is due to the angular differences for each band of each SCA and the timing differences between odd and even SCAs for when a given SCA acquires across-track geographic locations on the Earth surface. The SWIR-1 OLI band was used for calculating the angles shown in the figure.

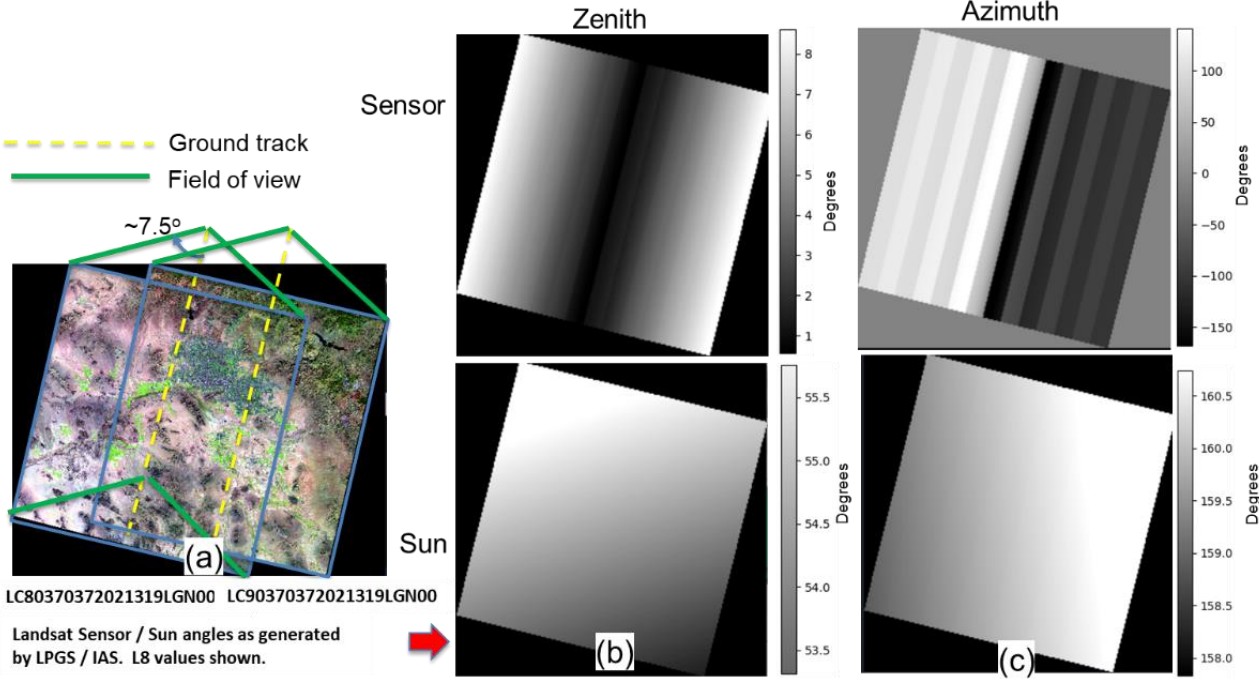

**Figure 9.** Landsat 9 and 8 images (**a**) during the underfly time-frame over WRS-2 path/row 037/037 acquired on 15 November 2021. The Landsat 9 sun and sensor viewing zenith (**b**) and azimuth (**c**) angles are also shown. Angle band images shown are contrast stretched to better show detail within each image. Angle band images are in units of degrees.

As the two acquisitions would not be viewing the same geographic location on the Earth's surface at the same time or with the same SCA or sensor angles, these SCA dependencies can cause differences in the two images' sun and sensor viewing angles. These differences can be difficult to visualize when looking at the data within a given scene. This is due to the fact that, when viewing the sun angles for a given acquisition for a single scene, as shown in Figures 9 and 10, using horizontal profiles of the data within a map-projected space in the angle band images, there is essentially a linear relationship across the acquisition with a small nonlinearity due to the viewing properties for each band of each SCA. To better show the nonlinear SCA-to-SCA dependencies for these sun angles for a given scene (WRS-2 037/037 acquired on 15 November 2021), the dominant linear relationship can be removed from the data, and the small angular differences between the SCAs can be shown for these sun angles. Figure 10 shows both these horizontal profiles derived from the data, providing a single profile for each angle, which is dominated by the linear relationship across the data, and then a profile with the linear component removed, demonstrating the SCA-to-SCA dependencies.

To demonstrate how these SCA-to-SCA dependencies drive differences in the sun and sensor viewing angles for the underfly image acquisitions, the sensor azimuth angles are shown in Figure 11 for path/row 037/037 acquired on 15 November 2021. This figure shows the sensor azimuth viewing angle bands for the two acquisitions projected to the output map projection space of the mosaic of the two images, the images of which were shown in Figure 9a, and the profiles at the same location for the mosaic are shown next to the sensor azimuth images for both acquisitions (Figure 11c). From Figure 11, it can be seen that the SCA number within the focal plane for Landsat 9 and 8 OLI instruments was generally not the same for the same location acquired in the overlap region. Figure 11 shows

that, although the viewing geometry was the same for both instruments of the images and for both acquisitions, the difference in their field of view coverage produced differences in the satellite viewing and correspondingly the sun angles for the two acquisitions in the overlap region. In many ways, as the display of the profiles show in Figure 11, however, the difference in these sensors' viewing azimuth angles can be visualized as a simple translation of the angles across the west-to-east direction of the images (although this is not exactly the case) from one acquisition field of view to the other. Although the differences in sensor azimuth angle, as seen in Figure 11, can be thought of as a simple shift between the two sensors, in reality, it is more complex due to the different SCAs viewing the geolocations on the ground at a slightly differing set of times.

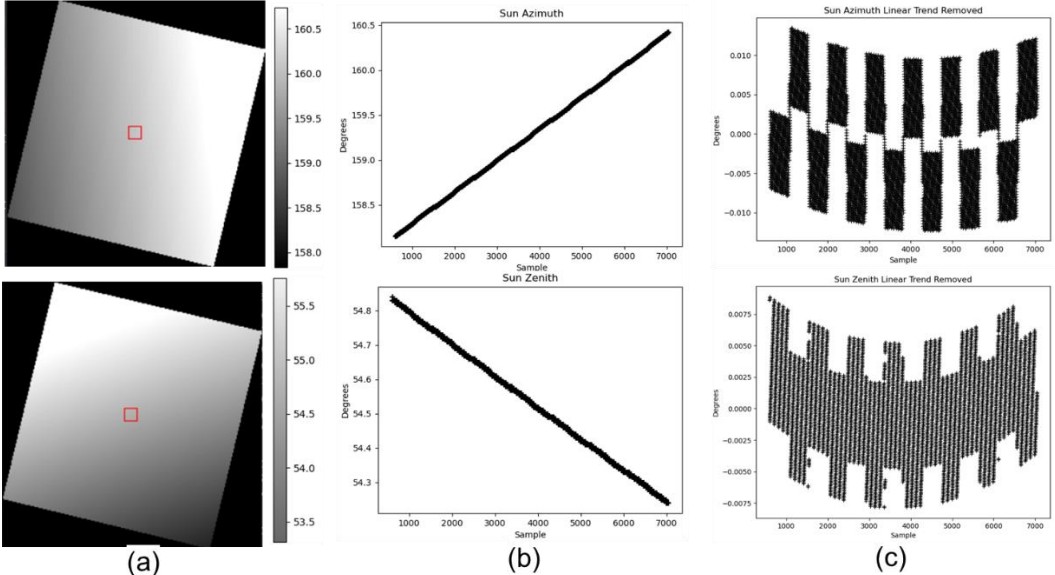

**Figure 10.** Sun angle across the Landsat 9 image over WRS-2 037/037 acquired on 15 November 2021. The far-left images (**a**) show the images as sun viewing angle bands, with the top images being the azimuth angle and the bottom being zenith angle. The center images (**b**) show a single horizontal profile of the imagery on the left, and the far-right images (**c**) show the profiles in the center with the linear trend removed from each profile plotted. The far-right profiles demonstrate the small SCA-to-SCA dependencies that show up in the sun viewing angles for the imagery. Angle band images shown are contrast stretched to better show detail within each image. Angle band images are in units of degrees.

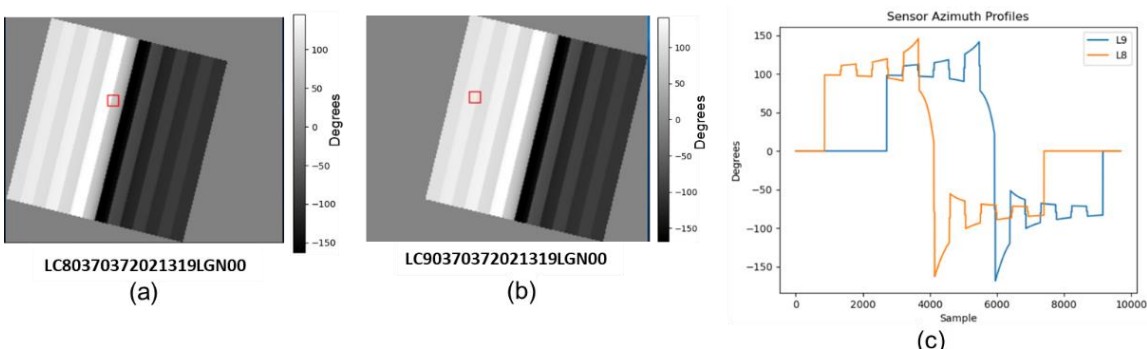

**Figure 11.** Sensor azimuth angles for two Landsat 9 and 8 acquisitions over WRS-2 037/037 acquired on 15 November 2021. Landsat 9 sensor viewing azimuth angles are shown in (**b**), while those for Landsat 8 are shown in (**a**). The plot in (**c**) shows a profile in the *y*-axis for the sensor azimuth angles images for (**a**,**b**). Angle band images shown are contrast-stretched to better show detail within each image. Angle band images are in units of degrees.

Figure 12 shows the sun and sensor viewing angle differences within the overlap region of the two acquisitions over WRS-2 037/037 acquired on 15 November 2021. Statistics for the original single scene sun and sensor viewing angles and the difference in the sun and sensor viewing angles are shown in Table 4. The image mosaic is shown in Figure 12a, while the difference in the azimuth sun and sensor viewing angles is shown in Figure 12b,c.

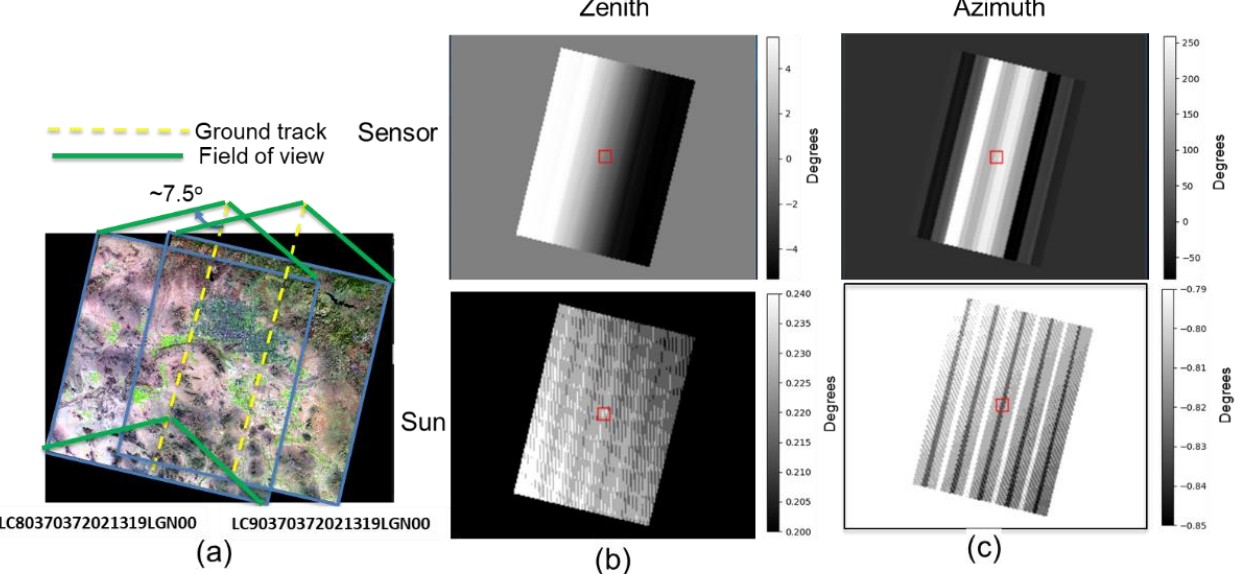

**Figure 12.** (**a**) Mosaic of the Landsat 9 and 8 underfly images over WRS-2 037/037 acquired on 15 November 2021. (**b**) Difference in the sensor and sun zenith angles between the two acquisitions in the region that has geographic overlap. (**c**) Difference in the sensor and sun azimuth angles between the two acquisitions in the region that has geographic overlap. Angle band images shown are contrast-stretched to better show detail within each image. Angle band images are in units of degrees.

**Table 4.** Sun (SUN) and sensor (SAT) viewing angle statistics for the Landsat 9 and 8 underfly image pair acquired over WRS-2 037/037 on 15 November 2021. Individual scene angles and the difference in angles are shown in the table. Angles are listed in units of degrees.

| Type | Direction | Standard Deviation | Mean | Maximum | Minimum |
|---|---|---|---|---|---|
| **L9** | SAT Azimuth | 96.4917 | 6.9863 | 141.7500 | −168.4600 |
| **L9** | SAT Zenith | 2.3523 | 4.4158 | 8.6100 | 0.5700 |
| **L9** | SUN Azimuth | 0.6459 | 159.2813 | 160.7400 | 157.8300 |
| **L9** | SUN Zenith | 0.5136 | 54.5334 | 55.7600 | 53.3100 |
| **L8** | SAT Azimuth | 96.8785 | 7.0721 | 145.7000 | −162.8400 |
| **L8** | SAT Zenith | 2.3569 | 4.4253 | 8.6500 | 0.5100 |
| **L8** | SUN Azimuth | 0.6533 | 159.4057 | 160.8800 | 157.9400 |
| **L8** | SUN Zenith | 0.5147 | 54.5010 | 55.7300 | 53.2800 |
| **Difference** | SAT Azimuth | 72.8360 | 28.3171 | 259.2000 | −80.3500 |
| **Difference** | SAT Zenith | 2.5077 | −0.0004 | 5.4100 | −5.3300 |
| **Difference** | SUN Azimuth | 0.3909 | −0.2866 | 0.0000 | −0.8500 |
| **Difference** | SUN Zenith | 0.1042 | 0.0763 | 0.2400 | 0.0000 |

The difference images in Figure 12 show the SCA-to-SCA dependencies and the differences that they cause between the two image's sun and sensor viewing angles. As shown in Table 4, the largest differences in magnitude occurred in the sensor azimuth angles. The differences in sun viewing angles were less than 1° between the two acquisitions.

Toward the end of the underfly time-frame, the Landsat 9 satellite was pointed off nadir during several descending passes to optimize the amount of geographic overlap between the L9 and L8 images, which would otherwise be minimal or even possibly nonexistent. On 16 November 2021, 57 descending rows were acquired as off-nadir viewing imagery. On

17 November 2021, 64 descending rows were acquired as off-nadir viewing imagery. These off-nadir angles ranged from −9.493° to −14.752°. This off-nadir pointing would impact the sun and sensor viewing angle geometry. Figure 13 shows an L9 and L8 underfly pair, where the L9 spacecraft was pointed off-nadir to cover a larger geographic area of the L8 field of view. These images were acquired over WRS-2 path/row 172/041 on 17 November 2021. The sensor viewing and sun angles per pixel are also shown in the figure for the L9 image. The L9 azimuth angles are the most noticeably different from a nominal nadir viewing acquisition. As the L9 spacecraft was pointed off-nadir, it introduced an angular bias in the sensor viewing angles, which removed the sharp transition in the azimuth angles at the center of the field of view, when compared with nadir looking acquisitions, and removed the peak in the sensor zenith angles. This is demonstrated by the angle band profiles taken from the two azimuth and zenith angles produced for each acquisition, as shown in Figure 14 and Figure 15, respectively.

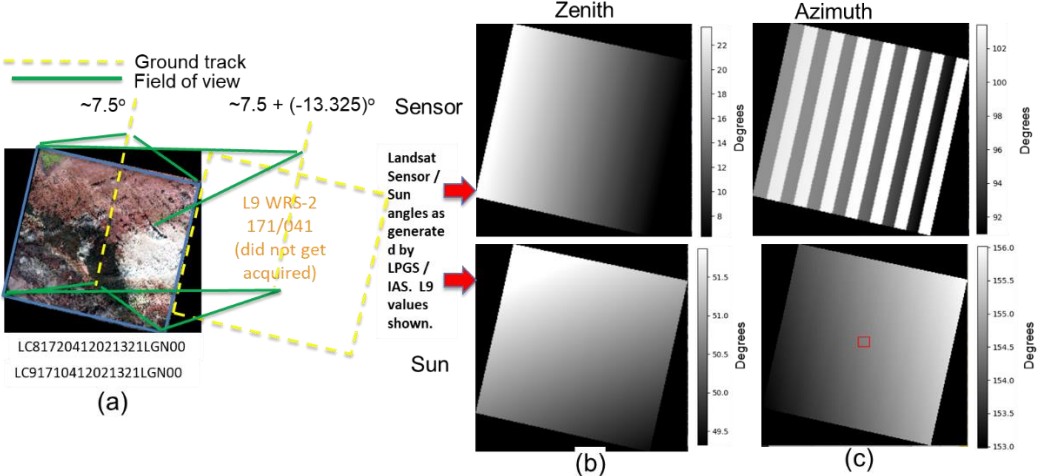

**Figure 13.** Landsat 9 (L9) and 8 (L8) images (**a**) acquired during the underfly time frame over WRS-2 path/row 172/041 acquired on 17 November 2021 and the Landsat 9 sun and sensor viewing zenith (**b**) and azimuth (**c**) angles. The Landsat 9 spacecraft was pointed off-nadir to improve the overlap between the Landsat images acquired. Angle band images shown are contrast-stretched to better show detail within each image. Angle band images in are units of degrees.

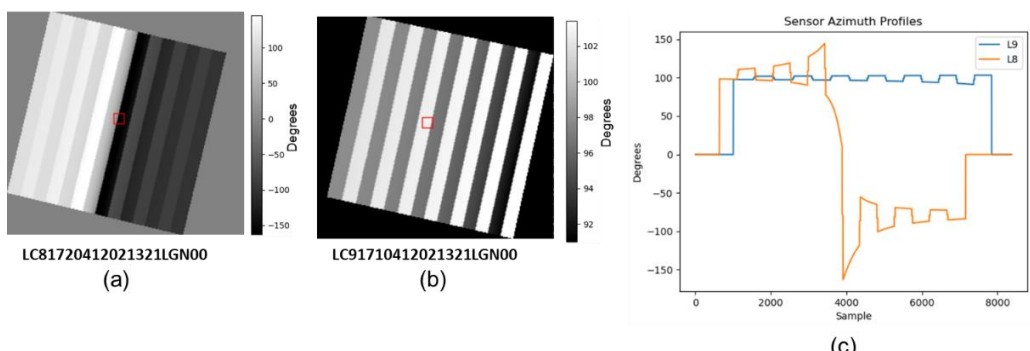

**Figure 14.** Sensor azimuth angles for two acquisitions over WRS-2 172/041 acquired on 17 November 2021 during the Landsat 9 (**b**) to Landsat 8 (**a**) underfly. The plot in (**c**) shows a profile in the *y*-axis for the sensor azimuth angles for the two acquisitions. The Landsat 9 spacecraft was pointed off-nadir by −13.325°. Angle band images shown are contrast=stretched to better show detail within each image. Angle band images are in units of degrees.

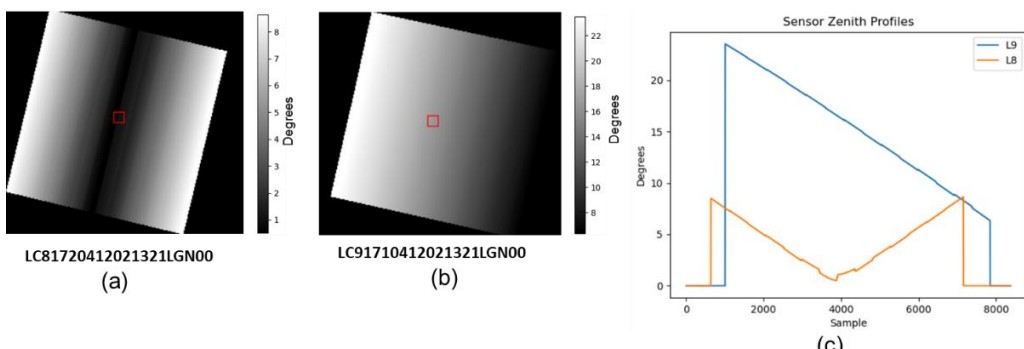

**Figure 15.** Sensor zenith angles for two acquisitions over WRS-2 172/041 acquired on 17 November 2021 during the Landsat 9 (**b**) to Landsat 8 (**a**) underfly. The plot in (**c**) shows a profile in the *y*-axis for the sensor azimuth angles for the two acquisitions. The Landsat 9 spacecraft was pointed off-nadir by $-13.325°$. Angle band images shown are contrast stretched to better show detail within each image. Angle band images are in units of degrees.

The sensor and sun viewing angle statistics in the overlap region for scenes shown in Figure 15 are shown in Table 5.

**Table 5.** Sun (SUN) and sensor (SAT) viewing angle statistics for the Landsat 9 and 8 underfly images acquired over WRS-2 172/041 on 17 November 2021. Individual scene angles and the difference in angles is shown in the table. Angles are listed in units of degrees.

| Type | Direction | Standard Deviation | Mean | Maximum | Minimum |
|------|-----------|--------------------|------|---------|---------|
| L9 | SAT Azimuth | 3.5546 | 99.1112 | 103.4100 | 91.0000 |
| L9 | SAT Zenith | 4.9548 | 15.1290 | 23.5300 | 6.3200 |
| L9 | SUN Azimuth | 0.6626 | 154.4958 | 156.0200 | 152.9800 |
| L9 | SUN Zenith | 0.5239 | 50.6002 | 51.8700 | 49.3200 |
| L8 | SAT Azimuth | 96.6132 | 6.6990 | 144.9500 | −162.8800 |
| L8 | SAT Zenith | 2.3572 | 4.4248 | 8.6500 | 0.5300 |
| L8 | SUN Azimuth | 0.6455 | 156.5703 | 158.0600 | 155.0800 |
| L8 | SUN Zenith | 0.5158 | 49.9218 | 51.1600 | 48.6800 |
| Difference | SAT Azimuth | 88.1228 | 56.8630 | 265.2500 | −48.0200 |
| Difference | SAT Zenith | 7.1289 | 6.9173 | 16.5800 | −0.4600 |
| Difference | SUN Azimuth | 1.1135 | −1.3194 | 0.0000 | −2.3200 |
| Difference | SUN Zenith | 0.3615 | 0.4281 | 0.7900 | 0.0000 |

## 4. Conclusions

During the commissioning of Landsat 9 and its ascent to its final WRS-2 orbit, the satellite was placed in an orbit that allowed it to underfly the L8 satellite and slowly drift across the L8 flight path, producing datasets that were near coincident in time between the two satellites and their sensors. Within this paper, the co-registration for the Level-1 precision and terrain-corrected imagery produced on the Landsat Product Generation System was shown to be within 2.2 m (RMSEr) across all bands for scenes with at least 10% overlap, less than 20% cloud cover within the scene, and at least 50% land. It was also estimated that, for images for which one of the image pairs failed precision correction and became an L1GT product type, a range of 8–14 m RMSEr could be expected in co-registration, while in cases where both image pairs failed the precision correction step and both images became an L1GT product type, a 14 m RMSEr could be expected for co-registration. With the image acquisition times being near-coincident, essentially removing temporal changes between the L9 and L8 image pairs, differences between the imagery would lie in the sun and sensor viewing angle differences between the acquisitions. The dependencies on these differences were shown for a nominal condition where both satellites were pointing nadir during the acquisition and where L9 was pointed off-nadir toward the end of the underfly period, which helped maximize the amount of overlap between the image pairs.

**Author Contributions:** Investigation, M.L.; Writing—original draft, M.J.C. and R.R.; Writing—review & editing, J.C.S. All authors have read and agreed to the published version of the manuscript.

**Funding:** Work performed under USGS contract G15PC00012.

**Data Availability Statement:** The Level-1 products discussed and used within the analysis of this paper can be accessed through Earth Explorer: https://earthexplorer.usgs.gov/ (accessed on 26 July 2022).

**Conflicts of Interest:** The authors declare no conflict of interest.

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
