# Peer review of "Landsat 9 Geometric Characteristics Using Underfly Data"

_remotesensing, doi:10.3390/rs14153781_

Round 1

Reviewer 1 Report

The Landsat 9 geometric accuracy is an important topic and the authors analyzed the results very well. The writing style is average and it (in particular some figures are rough) needs better clarification and writing to improve the impact and readiness of the paper.

Title – Could the title be changed to something like the Landsat 9 geometry characterization using underfly data?  Current title reads like this is only for Landsat-9 underfly period and the findings cannot be extended to normal period. Landsat-9 geometry in its non-underfly period is much more relevant to most users. Apparently, lines 235-238 state the conclusion applies to non-underfly period.

Abstract – should emphasize what was done and what were the findings in this study (currently only 5 out of 25 lines).

Figure 1, put an along track direction arrow in the figure would be very helpful. I understand this is quite straightforward for authors, but definitely not the case for general readers.

Figure 2 needs a lot clarification.

I don’t quite follow why staggered arrangement in Figure 1 will produce the saw tooth pattern in Figure 2 left (so called band 6 image). Landsat-8 is pushbroom sensor meaning the saw tooth pattern should not present in the image at all. Consider each SCA a painting brush (white), if I push such 14 staggered brushes to my floor, the white footprint should have a perfect linear edge without any saw tooth pattern.

Please explain clearly why the saw pattern appears (use a figure to illustrate if possible). In either case, I did not see such saw pattern in the L1TP product. Please explain what process has been done to get rid of such pattern if there is such a pattern in level 0 data.

Figure 2 band 6 is SWIR1 but the arrow in the figure pointing to SWIR2

The authors need specify why the time offset occurs. My guess is that the pointing direction difference between the odd and even SCA are different (one leaning forward and the other leaning backward along track direction). In other words, the time offset will be negligible if the odd and even SCA have no along track pointing direction differences. If this is the case, please specify this in the paper and specify why engineers make the odd and even SCA pointing differently (there must be a reason). If my guess is wrong, specify the reason.

Figure 2, “image lines are acquired at the same time” even confuses me more. Please clarify. Do the authors mean at different times?

The angle analysis is good. Just one question from the BRDF community. If I like to use one band viewing angles to represent all the 7 land band (blue, coastal, NIR, red, green, SWIR1 and SWIR2) angles to simply my analysis, would the red band the best candidate as it lies in the middle all the seven in Figure 1, i.e., its viewing angles are closer to other compared to the rest of the six bands? Maybe make this clear in the paper. The authors make some cool analysis on angles but links to BRDF community who urgently need the angle files is weak.

Line 83, is a detector corresponding to a pixel?

Section 2 briefly introduce the angle files and USGS I2I tasks. It should include the methods used in this study, i.e., how the underfly data are used to characterize the Landsat 9 geometry. For example, how Table 1 numbers are derived? Some correlation based registration quantification method or SIFT method.

In Table 2, add a column of process level such as L1GT and L1TP should be better.

Table 2 results are great. It makes a great difference to Landsat <=7, as the L1GT accuracy are still acceptable to the community. The L1GT Landsat-7 has much lower accuracy I believe. Put this in abstract.

Pre-fit Landsat-9 geometric accuracy is better than pre-fit Landsat 8. I noticed this result in Table 2 but the authors did not discuss this at all. What is the reason behind this?

Line 363 xx15.

Line 364-403, this long essay including Figure 8 need rewritten. I am totally lost on what the authors tried to say. I felt like they need a new paragraph, looks like they are disconnected with lines above 363.

Figure 9 and the following, which band is used for angle calculation, red band?

Figure 9 and the following, it is better to put the legend of the angle values.

Figure 10 (b), why the 0 angle values are present. I assume the authors also plot the filled values outside the scene. Please get rid of these values in Figure (b) so that the across swath angle difference are more evident.

Figure 10 (c), how many lines of profiles are plotted?

Figure 10 (c), the x-axis labels are weird. Why sample number (14000) can be larger than figure (b) and exceed normal ranges (7000-8000) of pixel width in a scene?

Line 522, off-nadir paragraph, important information, put this in abstract I guess.

Line 525 and 526, 57 rows. The unit of rows are wired (the L9 off-nadir in one row and then back to nadir in next row?). Is it path or path/row? If it is rows. Did the authors means continuous rows, e.g., from row 1 to row 57?

Line 527, Negative values are westward or eastward, this may be important for water sun glint and land BRDF community.

Author Response

Thank for reviewing the paper.  Your comments and suggestions were very helpful.  I have attached responses.  

Reviewer 2 Report

During the commissioning of Landsat 9 and its ascent to its final WRS-2 orbit, the satellite was placed in an orbit that allowed it to underfly the Landsat 8 satellite and slowly drift across the L8 flight path producing data sets that were near co-incident in time between the two satellites and their sensors. Then this paper describes some of the aspects of the underfly from an acquisition perspective, discusses the co-registration between the Level-1 products produced from the two systems, and describes the driving forces in the differences between sensor viewing and sun angles associated with the products, giving some examples of these differences. Within this paper these differences are only addressed for the multispectral data of the OLI instruments. This was a unique opportunity to perform such study. The results are useful and supported by the methodology used.

Some corrections and suggestions

L.36 - keyword 1; Landsat 8;  ---   Landsat 8; …

L.222- Landsat 8 scene scenter times  ---   Landsat 8 scenes center times

L.454- demonstrating the SCA-to-SCA dependences  ---  demonstrating the SCA-to-SCA dependencies

L.508, L.565 - Mininum ---  Minimum

Author Response

Thank you for reviewing the paper.  Your comments were welcomed and I appreciate the changes that you found that should be changed.  I have attached responses to your comments.

Reviewer 3 Report

Overall, the results of this paper will be useful to the Landsat community. However, more details and discussion need to be added to show how the results were reached. Specific comments to improve the manuscript are listed in the attached document.

Author Response

Thank your for reviewing the paper.  Your comments and suggestions will help with others understanding the content which is very much appreciated.

Round 2

Reviewer 3 Report

Thank you for answering the questions and comments that I posed to your original manuscript. I believe all my comments were addressed by version 2 of your manuscript and the necessary clarifications have been satisfied. One final suggestion is to state the units of the angles in the sun/sensor angle figures (Figure 12 for example) in the image caption (e.g.- "angle units are in degrees") in case it is not obvious to the reader.

Author Response

A statement of units for the angle band images was added to the captions.  Degrees was added as a label to angle band bar graphs / legends.  Thanks.